# SHARPIN Enhances Ferroptosis in Synovial Sarcoma Cells via NF-κB- and PRMT5-Mediated PGC1α Reduction

**DOI:** 10.3390/cancers15133484

**Published:** 2023-07-04

**Authors:** Hironari Tamiya, Naoko Urushihara, Kazuko Shizuma, Hisataka Ogawa, Sho Nakai, Toru Wakamatsu, Satoshi Takenaka, Shigeki Kakunaga

**Affiliations:** 1Department of Rehabilitation, Osaka International Cancer Institute, Osaka 541-8567, Japan; 2Department of Orthopaedic Surgery, Osaka International Cancer Institute, Osaka 541-8567, Japan; s.nakai.0925@gmail.com (S.N.); evolutionhhh49@yahoo.co.jp (T.W.); s.takenaka.0816@gmail.com (S.T.); shigeki.kakunaga@oici.jp (S.K.); 3Nitto Joint Research Department for Nucleic Acid Medicine, Research Center, Osaka International Cancer Institute, Osaka 541-8567, Japan; naoko.urushihara@nitto.com (N.U.); kazuko.shizuma@oici.jp (K.S.); hisataka.ogawa@oici.jp (H.O.)

**Keywords:** sarcoma, ferroptosis, TFRC, SHARPIN

## Abstract

**Simple Summary:**

Sarcoma is difficult to treat because of its rarity. Ferroptosis is a new type of cell death mediated by ferrous iron. In the present study, we aimed to clarify the effect of ferroptosis in sarcoma. As compared with noncancer and carcinoma cell lines, ferroptosis is more sensitive in most of the sarcoma cell lines. Moreover, transferrin receptor 1 (TFRC) and SHANK-associated RH domain interactor (SHARPIN), both of which are oncogenic factors and related to poor overall survival, are highly expressed, particularly in synovial sarcoma cell lines. Furthermore, we discovered that SHARPIN is a positive regulator of ferroptosis through nuclear factor-kappa B (NF-κΒ) and protein arginine methyltransferase 5 (PRMT5)-mediated PGC1α reduction. In summary, we suggest that ferroptosis could be a therapeutic target in sarcoma, particularly in subpopulations with high TFRC and SHARPIN expression.

**Abstract:**

Sarcoma is a rare type of cancer for which new therapeutic agents are required. Ferroptosis is a nonapoptotic cell death triggered by iron-mediated lipid peroxidation. We found that TFRC, an iron uptake protein, was expressed at higher levels in sarcoma cell lines than in noncancer and carcinoma cell lines. Glutathione peroxidase 4 (GPX4) protects cells against ferroptosis, and its inhibition using RAS-selective lethal 3 (RSL3) had an antitumor effect that was more pronounced in sarcoma cell lines, particularly synovial sarcoma cells, compared to non-sarcoma cells. Because NF-κB can provoke ferroptosis, we examined the role of SHARPIN, an activator of NF-κB, in sarcoma. We found that SHARPIN expression was significantly associated with reduced survival in cohorts of patients with cancer, including sarcoma. In addition, SHARPIN promoted the sensitivity of sarcoma cells to ferroptosis. Further analyses revealed that the PGC1α/NRF2/SLC7A11 axis and BNIP3L/NIX-mediated mitophagy are regulated through NF-κB and PRMT5 downstream of SHARPIN. Our findings suggest that ferroptosis could have a therapeutic effect in sarcoma, particularly in subpopulations with high TFRC and SHARPIN expression.

## 1. Introduction

Sarcoma is a rare type of cancer that accounts for less than 1% of all cancers. There are multiple sarcoma histotypes, including more than 100 types of soft tissue sarcomas [1]. The 5-year overall survival (OS) rate of patients with soft tissue sarcoma is around 70% [2]. Given the rarity and diversity of sarcomas, development of new therapeutic approaches is challenging. Despite the development of therapeutic agents such as pazopanib, trabectedin, and eribulin [3,4,5], treatment outcomes have remained unchanged over the past three decades [6,7]. Consequently, there is a need to identify new therapeutic targets for sarcoma.

Ferroptosis, a nonapoptotic form of cell death triggered by iron-dependent lipid peroxidation, was first discovered in 2012 [8] and is implicated as a mechanism of neurodegeneration in Parkinson’s and Alzheimer’s diseases as well as cancer and cardiovascular diseases and so on [9,10]. Iron is essential for DNA replication, DNA repair, and cell cycle control [11], but is a major catalyzer of reactive oxygen species’ (ROS) generation and can be toxic to cells in excessive amounts [12]. Uptake of iron is upregulated in cancer cells [13,14], which seem to take advantage of the oncogenic aspects of iron while evading its toxic effects. Several studies have suggested that the induction of ferroptosis could be used as a treatment for cancer [15,16,17]. Little is known about the potential therapeutic effects of ferroptosis in sarcoma; however, mesenchymal tumors are more sensitive to ferroptosis than epithelial tumors [18]. Nonetheless, the precise mechanism underlying this sensitivity has not been fully elucidated.

Nuclear factor-kappa B (NF-κΒ) and protein arginine methyltransferase 5 (PRMT5) function as oncogenes in many types of cancer [19]. SHANK-associated RH domain-interacting protein (SHARPIN), a subunit of the linear ubiquitin assembly complex (LUBAC), is involved in NF-κΒ signaling and apoptosis [20]. In addition, SHARPIN binds directly to PRMT5 to enhance its activity [21,22] and this interaction promotes the growth of melanoma (Tamiya et al., 2018). SHARPIN expression is associated with poor prognosis in some other types of cancer; for example, SHARPIN-mediated activation of NF-κΒ signaling and the downstream targets survivin and livin promotes the progression and metastasis of prostate cancer [23]. In addition, overexpression of PRMT5 is related to the poor prognosis of multiple myeloma and has been described as a druggable target [24].

The potential roles of NF-κΒ and PRMT5 in the development of sarcoma have not been elucidated. A recent study demonstrated that NF-κΒ activation is required to trigger ferroptosis [25]; therefore, we investigated the influence of SHARPIN on the prognosis of sarcoma and the sensitivity of sarcoma cells to ferroptosis. Our findings highlight the potential use of ferroptosis as a therapeutic approach in sarcoma. 

## 2. Materials and Methods

### 2.1. Animal Studies

For xenograft experiments, Yamato-SS (Yamato) synovial sarcoma cells (1.0 × 10^7^) were injected subcutaneously into the lower right flank of 4-week-old female nude mice (The Jackson Laboratory Japan, Yokohama, Japan). RAS-selective lethal 3 (RSL3) treatment was initiated on day 11 (3 days after the mean tumor size reached 50 mm^3)^. Intratumoral injection of RSL3 (30 mg/kg in vehicle comprising 2% DMSO, 30% PEG300, 2% Tween-80, and 66% sterile water) was conducted three times per week. Tumor sizes were monitored using calipers up to 27 days after injection. Tumor volume was calculated as (A × B^2^)/2, where A is the longest diameter and B is the shortest diameter. Experimental scheme is shown below.

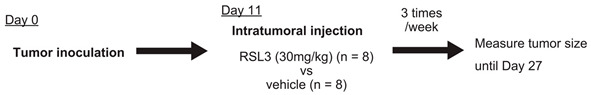


### 2.2. Cell Culture

All cell lines were cultured in DMEM (Gibco, Waltham, MA, USA) containing 10% FBS (Gibco), 100 IU/mL penicillin, and 100 µg/mL streptomycin (Nacalai Tesque, Kyoto, Japan). The cells were maintained in the growth phase and did not exceed 80% confluency. HDF is an immortalized human dermal fibroblast cell line. HEK293T (293T) cells are human embryonic kidney cells. Hela is a cervical carcinoma cell line. The Hela cell line is one of the most common carcinoma cell lines. HDF and HEK293T are also universal cell lines. Taking universality of these three cell lines into consideration, they can be regarded as representative non-sarcoma cell lines in the present study. SU-CCS-1 (SU), KAS, MP-CCS-SY (MP), and Hewga-CCS (Hewga) are clear cell sarcoma (CCS) cell lines. NEPS and VAESBJ are epithelioid sarcoma cell lines. U2OS, 143B, HOS, Saos-2, and MG63 are osteosarcoma cell lines. Aska-SS (Aska), Yamato, SYO-1, and HS-SY-II are synovial sarcoma cell lines. EW8, HT-1080, and SW872 are Ewing sarcoma, fibrosarcoma, and liposarcoma cell lines, respectively. 

### 2.3. Antibodies and Reagents 

Antibodies targeting the following proteins were used: TFRC (ab214039, Abcam, Cambridge, UK), FPN (NBP1-21502, Novus Biologicals, Englewood, CO, USA), FTH1 (ab65080, Abcam), SLC7A11 (600-401-GU3, Rockland Immunochemicals, Pottstown, PA, USA), GPX4 (ab125066, Abcam), GAPDH (sc-32233, Santa Cruz Biotechnology, Dallas, TX, USA), PGC1α (NBP1-04676, Novus Biologicals), SHARPIN (ABF128, Millipore, Burlington, MA, USA), β-actin (#4970, Cell Signaling Technology, Danvers, MA, USA), VDAC1/3 (ab14734, Abcam), PRMT5 (sc-376937, Santa Cruz Biotechnology), SDMA (SYM10; 07-412, Millipore), SOX10 (sc-365692, Santa Cruz Biotechnology), MITF (ab12039, Abcam), LC3B (NB100-2220, Novus Biologicals), NRF2 (#12721, Cell Signaling Technology), Parkin (#4211, Cell Signaling Technology), and BNIP3L/NIX (#12396, Cell Signaling Technology). For NRF2 detection, samples were pretreated with 50 μM MG132 (FUJIFILM Wako Pure Chemical Corporation, San Diego, CA, USA) for 3 h. Complex I/III/V was detected using Total OXPHOS Human WB Antibody Cocktail (ab110411, Abcam). Erastin, EPZ015666, ATN-161, and RGD peptide were purchased from Selleck Chemicals (Houston, TX, USA). TGF-β was purchased from BioLegend (San Diego, CA, USA). RSL3, SC-514, and ferrostatin-1 were purchased from MedChemExpress (Monmouth Junction, NJ, USA). Deferoxamine was obtained from Cayman Chemicals (Ann Arbor, MI, USA).

### 2.4. Immunoblotting

Cells were lysed by incubation for 20 min at 4 °C in TBS-lysis buffer (comprising TBS (50 mM Tris-HCl, pH 7.5, 150 mM NaCl) supplemented with 1% (*v*/*v*) Triton X-100 and 1× ProteoGuard EDTA-Free Protease Inhibitor Cocktail (Takara Bio, Kusatsu, Japan)). The supernatant was collected after centrifuging the sample at 15,000 rcf for 20 min at 4 °C and was then boiled in Laemmli buffer before separation by SDS-PAGE and transferal to a PVDF membrane. Membranes were incubated for 1 h at room temperature with blocking solution (TBS containing 0.1% Tween-20 and 5% nonfat milk) and then overnight at 4 °C with the appropriate primary antibody. Subsequently, the membranes were washed with TBS and incubated for 1 h at room temperature with the secondary antibody (HRP-conjugated goat antirabbit or goat antimouse antibody) (Bio-Rad, Hercules, CA, USA). Protein bands were visualized and quantified using Clarity Western ECL Substrate (Bio-Rad) and the ImageQuant LAS 4000 Imaging System (Fujifilm).

### 2.5. Gene Silencing

SHARPIN-specific shRNA lentiviral vectors with a pLKO.1 backbone were gifted by the Ronai laboratory (Sanford Burnham Prebys Medical Discovery Institute, La Jolla, CA, USA). Lentiviral particles were prepared using standard protocols. Briefly, the shRNA plasmid and the second-generation packaging plasmids ΔR8.2 and VSV-G (Addgene, Watertown, MA, USA) were transfected into 293T cells. Viral supernatants were collected after 48 h of culture and used with polybrene (Santa Cruz Biotechnology) for infection. For SHARPIN RNAi, cells were transfected with an siRNA targeting SHARPIN or with a corresponding scrambled siRNA (final siRNA concentration: 10 nM) (SMARTpool reagents, Horizon Discovery) using Lipofectamine RNAiMAX Transfection Reagent (Thermo Fisher Scientific, Waltham, MA, USA).

### 2.6. RNA Extraction and qPCR

Total RNA was isolated from cells using the RNeasy Mini Kit (Qiagen, Hilden, Germany) and reverse transcribed using SuperScript IV VILO (Thermo Fisher Scientific). RNAlater solution (Thermo Fisher Scientific) was used to stabilize RNA in clinical samples.

The qPCR analyses were performed with a StepOnePlus Real-Time PCR System (Applied Biosystems, Waltham, MA, USA) using iTaq Universal SYBR Green Supermix (Bio-Rad). GAPDH was amplified as a control. Sequence-specific primers are shown in Appendix A.

### 2.7. Cell Viability/Cell Death Assays

Cells were seeded into 96-well plates at a density of 5.0 × 10^3^ cells/well and incubated with RSL3 (or vehicle) for 24 h. Subsequently, cell viability was quantified using Cell Counting Kit-8 (Dojindo, Kumamoto, Janpan) and an Infinite 200 PRO M Plex plate reader (Tecan, Kanagawa, Japan).

### 2.8. Immunohistochemistry

Formalin-fixed, paraffin-embedded, microtomed tumor sections were hydrated by passage through xylene and graded ethanol solutions. After antigen retrieval by incubation for 10 min at 99 °C in citric buffer (pH 6.0) (all antibodies other than anti-TFRC) or in Tris/EDTA buffer (pH 9.0) (anti-TFRC), the slides were blocked with 1% BSA and 10% FBS/PBS for 1 h and then incubated with a primary antibody for 16 h at 4 °C. Subsequently, the slides were washed with PBS, mounted using ImmPRESS Reagent (Vector Laboratories, Burlingame, CA, USA), and counterstained with hematoxylin.

### 2.9. Detection of Lipid Peroxidation

Yamato cell lines were seeded into 6-well plates and incubated with 5 µM BODIPY 581/591 C11 (Invitrogen, Waltham, MA, USA) for 30 min at 37 °C. Next, 1 µM RSL-3 was added for the indicated time, and green fluorescence was detected as lipid peroxidation using a FLUOVIEW FV10i confocal laser-scanning microscope (Olympus, Tokyo, Japan).

### 2.10. Complex I Activity Assay

Lysates were prepared from 1.0 × 10^7^ Yamato cells with or without SHARPIN knockdown. The protein concentration of each lysate was measured and standardized. Complex I activity was measured in each well at OD 450 nm using an Infinite 200 PRO M Plex plate reader (Tecan). Measurements were performed according to the manufacturer’s protocol.

### 2.11. ROS Assay

Yamato cells (1.0 × 10^4^) with or without SHARPIN knockdown were seeded into 96-well plates and incubated overnight. The cells were treated with RSL3 as indicated in the figure legends, and ROS levels were determined using the ROS Assay Kit-Highly Sensitive DCFH-DA-(Dojindo) according to the manufacturer’s protocol.

### 2.12. GSH/GSSG Ratio Assay

Yamato cells (1.0 × 10^7^) with or without SHARPIN knockdown were collected, and the concentration of GSH/GSSG was measured using a GSSG/GSH Quantification Kit (Dojindo) according to the manufacturer’s protocol.

### 2.13. In-Silico Analysis

All data from The Cancer Genome Atlas (TCGA) were obtained using cBioportal for Cancer Genomics. The TCGA database was analyzed according to a certain gene amplification and/or mRNA upregulation to clarify the association with survival. The Cancer Cell Line Encyclopedia (CCLE) was analyzed for RPPA signal and GPX4 dependency.

### 2.14. Statistics

All data are expressed as the mean ± SD unless otherwise specified. Group differences were analyzed using Student’s *t*-tests (two-tailed). Statistical analyses of multiple groups were performed using one- or two-way ANOVA. SHARPIN mRNA expression levels in normal tissue and synovial sarcoma or CCS samples are expressed as the median ± interquartile range. Differences were analyzed using a Mann–Whitney U test (two-tailed). In xenograft experiments, tumor volumes were analyzed using two-way ANOVA. Patient survival data were analyzed using a log-rank test. *p* ≤ 0.05 was considered statistically significant. All statistical analyses were performed using EZR (64-bit) [26].

## 3. Results

### 3.1. Sarcoma Cell Lines Are More Sensitive to Ferroptosis than Noncancer or Carcinoma Cell Lines

Expression of transferrin receptor 1 (TFRC), which is essential for iron uptake, is associated with increased sensitivity of cells to ferroptosis [27,28,29]. First, we used quantitative PCR (qPCR) to compare the expression levels of the TFRC mRNA in noncancer, carcinoma, and sarcoma cell lines. TFRC mRNA was more abundant in most sarcoma cell lines than in noncancer or carcinoma cell lines (Figure 1A). Moreover, analysis of a Cancer Cell Line Encyclopedia (CCLE) dataset [30] revealed that TFRC protein expression was high in soft tissue sarcoma cell lines but relatively low in bone cancer cell lines (Figure 1B).

Solute carrier family 7 member 11 (SLC7A11) and glutathione peroxidase 4 (GPX4) are key regulators that protect cells against ferroptosis [31,32,33]. We examined the effects of erastin and RAS-selective lethal 3 (RSL3), inhibitors of SLC7A11 and GPX4, respectively, on the sensitivities of several cell lines to ferroptosis. RSL3 had a strong cytotoxic effect on most sarcoma cell lines tested but had a much less pronounced cytotoxic effect on human dermal fibroblasts (HDF) and HEK293T (293T) cells (Appendix A). With the exception of that for the U2OS osteosarcoma cell line, the half maximal inhibitory concentrations (IC50) of RSL3 for sarcoma cell lines were lower than those for HDF, 293T cells, and HeLa cells (Figure 1C). Erastin, an inhibitor of the cystine–glutamate antiporter SLC7A11, had a strong cytotoxic effect on sarcoma cell lines as well as noncancer and carcinoma cell lines (Appendix A). 

Next, we examined the expression levels of TFRC and other proteins relevant to iron metabolism, including ferroportin (FPN), which is an iron export regulator, and ferritin heavy chain (FTH1), which is a subunit of the major iron storage protein ferritin. In line with the qPCR data (Figure 1A), an immunoblot analysis revealed that TFRC protein expression was higher in the KAS, NEPS, U2OS, 143B, Aska, Yamato, and HT1080 sarcoma cell lines than in HDF, 293T cells, and HeLa cells (Figure 1D). The SU, VAESBJ, and EW8 sarcoma cell lines expressed lower levels of TFRC than the aforementioned sarcoma cell lines, whereas FPN was expressed at a high level in 293T and 143B cells (Figure 1D). FTH1 expression was high in some ferroptosis-resistant cells, including HDF and 293T cells (Figure 1D); this finding is in line with a previous study showing that increased expression of FTH1 reduces labile iron and ultimately suppresses cell death induced by ferroptosis [34]. The SLC7A11 protein was expressed at a high level in HDF, 293T cells, and U2OS sarcoma cells while the expression of GPX4 protein was high in HDF, Aska and Yamato cell lines (Figure 1D). The LC3B, a marker of autophagic activity that is associated with ferroptosis sensitivity through ferritinophagy [34], varies depending on the cell lines; HDF, Aska, Yamato, HT1080 and SW872 cell lines highly expressed LC3B (Figure 1D). 

Using the CCLE, we analyzed data from Chronos, an algorithm for estimating the fitness effects of gene knockout in CRISPR screens [35,36,37]. This analysis revealed that bone cancer and soft tissue sarcoma cell lines have high GPX4 dependency (Figure 1E).

### 3.2. Synovial Sarcoma Cells Are Sensitive to Ferroptosis Inducers and Express High Levels of Iron Metabolism-Related Proteins

Next, we investigated the sensitivity of various sarcoma cell lines to ferroptosis induced by RSL3. All four synovial sarcoma cell lines examined were extremely sensitive to ferroptosis (IC50: SYO-1, 0.014 μM; HS-SY-II, 0.020 μM; Yamato, 0.046 μM; Aska, 0.0088 μM), whereas osteosarcoma cell lines were relatively resistant to RSL3 (IC50: U2OS, 9.731 μM; Saos-2, 4.426 μM; MG63, 4.994 μM; HOS, 1.481 μM; 143B 0.237 μM) (Figure 2A). Erastin treatment confirmed the high sensitivity of the Aska and Yamato cell lines to ferroptosis (Appendix A). Consequently, we focused on ferroptosis-sensitive synovial sarcoma cells in subsequent experiments. We treated the Aska and Yamato cell lines with a lower concentration of RSL3 and with or without an inhibitor of ferroptosis (deferoxamine (DFO) or ferrostatin-1 (Fer-1)). DFO and Fer-1 both blocked the cytotoxic effect of RSL3 in Aska and Yamato cells (Figure 2B,C), confirming that this effect was indeed attributable to the induction of ferroptosis. RSL3 treatment also induced lipid peroxidation in Yamato cells (Figure 2D).

To clarify the association between iron metabolism and the prognosis of sarcoma in clinical practice, we investigated the relationship between OS and the expression levels of the TFRC, FPN, and FTH1 genes in patient samples. First, we analyzed existing data for 10,967 samples in The Cancer Genome Atlas (TCGA) database [38,39], representing all types of cancers. TFRC gene amplification was significantly associated with shorter OS (*p* < 0.001) and disease-free survival (DFS) (*p* < 0.001) (Figure 2E,F), whereas FPN and FTH1 gene alterations had no associations with these parameters (Appendix A). Furthermore, an analysis of patients with synovial sarcoma treated in our institute revealed a significant correlation between high TFRC mRNA expression and poor OS for both the full cohort (n = 39, *p* < 0.005) and the cohort of patients with advanced-stage disease (n = 13, *p* < 0.05) (Appendix A). By contrast, we did not identify a significant correlation between FTH1 expression and OS for either the full cohort (n = 39) or the subset of patients with localized synovial sarcoma (n = 26) (Appendix A). Immunostaining of eight paraffin-embedded synovial sarcoma specimens revealed that seven specimens were positive for TFRC, and all eight specimens were positive for FTH1 (Figure 2G), suggesting abundant iron uptake and storage in synovial sarcoma cells. 

Next, we evaluated the antitumor effect of RSL3 in vivo. The Yamato synovial sarcoma cell line was inoculated into nude mice, and RSL3 (or vehicle) was injected intratumorally. Compared with the vehicle control, RSL3 reduced tumor progression significantly (*p* < 0.0005) (Figure 2H).

Overall, the data described above suggests that the induction of ferroptosis could have a promising therapeutic effect in synovial sarcoma, and that high TFRC expression is associated with poor prognosis of the disease.

### 3.3. SHARPIN Is Expressed at a High Level in Synovial Sarcoma and Is Associated with Poor Prognosis of the Disease

As an activator of NF-κΒ, SHARPIN plays a role in cancer progression [40,41]; therefore, we hypothesized that it may impact the prognosis of sarcoma. First, we examined the expression of SHARPIN in several cell lines. With the exception of VAESBJ (epithelioid sarcoma), HT1080 (fibrosarcoma), and SW872 (liposarcoma) cells, the expression level of the SHARPIN mRNA in the other sarcoma cell lines was higher than that in HDF, although it was also expressed at a high level in 293T and HeLa cells (Figure 3A). Notably, the SHARPIN mRNA level in the four synovial sarcoma cell lines was more than twice as high as that in HDF. In addition, analyses of clinical samples revealed that SHARPIN mRNA expression was significantly higher (*p* < 0.005) in synovial sarcoma (n = 40) than in normal tissue (n = 10) (Figure 3B).

Next, we analyzed the relationship between SHARPIN expression and patient survival using existing data for 10,967 samples in the TCGA database, including all types of cancer. In this analysis, SHARPIN gene amplification was significantly associated with shorter progression-free survival (PFS) (*p* < 0.001) and DFS (*p* < 0.001) but was not associated with OS or disease-specific survival (DSS) (Figure 3C,D and Appendix A). By contrast, in another cohort of samples (ICGC/TCGA Pan-Cancer Analysis of Whole Genomes Consortium, 2020; n = 362), including all types of cancer, SHARPIN gene amplification and/or high mRNA expression was associated with shorter OS (Figure 3E). Analysis of a sarcoma-specific cohort (n = 265) revealed that SHARPIN gene amplification and/or high mRNA expression was significantly associated with shorter OS (*p* < 0.05) and PFS (*p* < 0.05) (Figure 3F,G). Furthermore, an analysis of synovial sarcoma samples from patients treated in our institute revealed that SHARPIN mRNA expression was significantly associated with shorter PFS (*p* < 0.005) but was not associated with OS (Figure 3H and Appendix A). A subsequent analysis of a TCGA dataset of soft tissue sarcoma samples (Firehose Legacy, n = 251) revealed that the mRNA expression levels of the other LUBAC components (HOIP and HOIL-1L) and those of PRMT5, the activity of which is modulated by SHARPIN, were not significantly associated with OS (Appendix A). Taken together, these findings highlight the impact of SHARPIN on clinical outcomes such as OS and DFS, particularly in sarcoma.

### 3.4. SHARPIN Promotes the Sensitivity of Synovial Sarcoma Cell Lines to Ferroptosis by Inhibiting the PGC1α/SLC7A11 Axis

The results described above suggested that TFRC and SHARPIN can impact the prognosis of all types of cancer and sarcoma specifically. TFRC increases the sensitivity of cells to ferroptosis [27,28], and NF-κΒ activation is required to trigger ferroptosis [25].

To elucidate the role of SHARPIN, an activator of NF-κΒ, on ferroptosis, we performed a knockdown experiment in synovial sarcoma cell lines. Knockdown of SHARPIN enhanced the resistance of the Yamato and Aska cell lines to ferroptosis induced by RSL2 or erastin (Figure 4A–D), suggesting that SHARPIN promotes ferroptosis sensitivity. To identify the mechanism underlying this role of SHARPIN, we examined protein expression levels of the ferroptosis-related factors TFRC, GPX4, and SLC7A11 in SHARPIN knockdown Yamato and Aska cells. Knockdown of SHARPIN did not affect TFRC or GPX4 protein levels in either cell line (Appendix A) but did upregulate SLC7A11 protein and mRNA levels in Yamato (Figure 4E,F) and its protein levels in Aska (Figure 4E) cell lines. In addition, knockdown of SHARPIN increased the protein levels of peroxisome proliferator-activated receptor-gamma coactivator 1α (PGC1α) and nuclear factor erythroid 2–related factor 2 (NRF2) (Figure 4E,G), both of which control SLC7A11 expression [42,43]. These results suggest that SLC7A11 expression is controlled at the mRNA level downstream of SHARPIN.

PGC1α regulates mitochondrial biogenesis [44]; therefore, we examined the effects of knockdown of SHARPIN on the levels of oxidative phosphorylation (OXPHOS)-related factors, including complexes I, III, and V, in Yamato cells. Knockdown of SHARPIN increased the level of complex I protein but had no effect on those of the complex III and V proteins (Figure 4E). In addition, knockdown of SHARPIN in Yamato cells increased the activity of complex I (Figure 4H). The protein expression level of voltage-dependent anion channel 1/3 (VDAC1/3), a key protein that regulates mitochondrial function, was not affected by knockdown of SHARPIN in Yamato cells (Figure 4E). Because mitochondrial function was enhanced by knockdown of SHARPIN without the induction of VDAC1/3, we focused on the mitophagy pathway. Further investigation revealed that gene silencing of SHARPIN did not affect Parkin-dependent mitophagy but induced adenovirus E1B 19-kDa-interacting protein 3-like (BNIP3L/NIX)-mediated mitophagy (Figure 4E), which is characterized by the induction of BNIP3L/NIX and microtubule-associated proteins 1A/1B light chain 3B (LC3B) [45].

OXPHOS can enhance the generation of ROS [46]; therefore, we examined whether knockdown of SHARPIN increases intracellular ROS production through the induction of OXPHOS. Under basal conditions, the ROS level in Yamato cells expressing a SHARPIN-specific shRNA was comparable to that in cells expressing a scrambled shRNA; however, following treatment with RSL3, ROS induction was suppressed in SHARPIN knockdown cells versus control cells (Figure 4I). In addition, SHARPIN knockdown cells exhibited a higher GSH/GSSG ratio, an indicator of oxidative stress (Figure 4J), suggesting higher antioxidant defense. According to the CCLE database, high SHARPIN mRNA expression is significantly associated (High vs Middle, *p* < 0.005; High vs Low, *p* < 0.05) with high GPX4 dependency in a bone and soft tissue sarcoma cohort (Figure 4K). Overall, these findings suggest that SHARPIN enhances the sensitivity of cells to ferroptosis by inhibiting the PGC1α/NRF2/SLC7A11 axis and BNIP3L/NIX-mediated mitophagy, both of which play a role in the antioxidant response against ROS.

### 3.5. Clear Cell Sarcoma Cells Expressing High Levels of PGC1α Do Not Respond to SHARPIN-Mediated Regulation of Ferroptosis Sensitivity

Given that the results above identified a role of PGC1α in controlling cellular sensitivity to ferroptosis, we measured PGC1α mRNA expression in several sarcoma and non-sarcoma cell lines. The PGC1α mRNA level was higher in two clear cell sarcoma (CCS) cell lines (SU and KAS) than in the other sarcoma and non-sarcoma cells (Figure 5A). We also examined the protein levels of SHARPIN, PRMT5, symmetric dimethylarginine (SDMA), the production of which is catalyzed by PRMT5, the melanocytic markers SRY-box transcription factor 10 (SOX10) and microphthalmia-associated transcription factor (MITF), PGC1α, and SLC7A11 in CCS cell lines and HDF. SHARPIN protein expression was substantially higher in CCS cell lines than in HDF (Figure 5B). Using clinical samples, we confirmed that SHARPIN mRNA levels were significantly higher (*p* < 0.05) in CCS samples than in normal tissue (Figure 5C). PRMT5 expression was comparable between CCS cell lines and HDF (Figure 5B). Expression of SDMA varied depending on the cell line, whereas SOX10 and MITF were expressed in all CCS cell lines, especially SU and KAS, but were not expressed in HDF (Figure 5B). The anti-PGC1α antibody identified post-translationally modified proteins, as described previously [47,48]. PGC1α was expressed in HDF, but bands representing modified PGC1α were observed only in CCS cell lines. Consistently, SLC7A11 expression was higher in CCS cell lines than in HDF (Figure 5B).

MITF regulates the expression of PGC1α [49]. According to a previous study, EWS-ATF1 fusion genes constitutively activate MITF transcription in CCS [50]; therefore, we hypothesized that EWS-ATF1-induced overexpression of MITF mediates PGC1α induction in CCS cells and overwhelms the inhibitory effect of SHARPIN on PGC1α expression. As expected, knockdown of SHARPIN did not impact PGC1α protein expression in CCS cell lines (Figure 5D). In addition, knockdown of SHARPIN did not alter the sensitivity of CCS cell lines to ferroptosis (Figure 5E). Therefore, we concluded that PGC1α is a key factor involved in SHARPIN-mediated regulation of cellular sensitivity to ferroptosis.

### 3.6. NF-κΒ and PRMT5 Are Involved in the Regulation of Sensitivity to Ferroptosis by SHARPIN

Next, we investigated the mechanism by which SHARPIN alters ferroptosis sensitivity through the PGC1α/SLC7A11 axis. SHARPIN controls the activities of NF-κΒ and PRMT5 [20,21], which regulate the production of ROS and are involved in both apoptotic and nonapoptotic cell death [51,52]; hence, we analyzed whether knockdown of SHARPIN inhibits the activities of NF-κΒ and PRMT5 in synovial sarcoma (Yamato) cells. Immunoblotting and qPCR analyses showed that inhibition of SHARPIN downregulated the levels of SDMA protein and IL-6 mRNA (a NF-κΒ target gene), indicating that SHARPIN controls the activities of PRMT5 and NF-κΒ in Yamato cells (Figure 6A). Inhibition of PRMT5 (using EPZ015666) increased the protein levels of PGC1α and SLC7A11 and suppressed that of SDMA in Yamato cells (Figure 6B). Moreover, inhibition of NF-κΒ (using SC-514) also increased the protein levels of PGC1α and SLC7A11 in Yamato cells (Figure 6C). Other SHARPIN-related factors, including integrin and transforming growth factor-β (TGF-β [21,53], did not affect the protein expression levels of PGC1α or SLC7A11 in Yamato cells (Appendix A).

Analysis of a TCGA dataset including all types of cancer suggested that amplification of either the SHARPIN gene or the TFRC gene is associated with DFS (Figure 6D). In addition, amplification of both genes is significantly associated with a significantly shorter DFS (no amplification (AMP) group, n = 4824, median DFS: N/A, 95% CI: 209.8–N/A; SHARPIN only group, n = 277, median DFS: 66.1, 95% CI: 52.0–101.1; TFRC only group, n = 245, median DFS: 96.1, 95% CI: 61.6–N/A; SHARPIN and TFRC group, n = 37, median DFS: 35.1, 95% CI: 22.8–N/A, *p* < 0.001) (Figure 6D). Similarly, analysis of a TCGA dataset of sarcoma samples revealed that gene amplification or high mRNA expression of SHARPIN and/or TFRC is associated with a significantly shorter OS (no alteration group, n = 207, median OS: 76.4, 95% CI: 54.2–88.5; SHARPIN amplification or HIGH group, n = 15, median OS: 26.4, 95% CI: 9.3–N/A; TFRC AMP or HIGH group, n = 15, median OS: 65.4, 95% CI: 13.6–N/A; SHARPIN and TFRC AMP or HIGH group, n = 4, median OS: 3.88, 95% CI: 0.6–N/A, *p* < 0.05) (Figure 6E). 

## 4. Discussion

Based on the results presented above, we propose that sarcoma is characterized by increased expression of TFRC and SHARPIN, with high GPX4 dependency, suggesting that ferroptosis may be a promising therapeutic target for sarcoma. Our initial experiment showed that TFRC was expressed at a high level in most sarcoma cell lines examined. TFRC is upregulated in various types of cancer [54,55]; although little is known about the role of TFRC in sarcoma, a recent study showed that it is associated with poor prognosis of osteosarcoma [56]. Here, analysis of a TCGA dataset including all types of cancer revealed that TFRC was significantly associated with poor prognosis (Figure 2E,F). A similar finding was obtained when analyzing data for synovial sarcoma patients treated in our institute (Appendix A), although the sample number was too small to reach a definitive conclusion about the role of TFRC in sarcoma. In addition, analysis of a CCLE dataset revealed upregulation of the TFRC protein level in soft tissue sarcoma cell lines (Figure 1B). Furthermore, seven of eight synovial sarcoma clinical samples were positive for TFRC expression, and all eight were positive for FTH1 expression. Taken together, these findings suggest that iron metabolism plays an important role in both carcinoma and sarcoma, particularly synovial sarcoma, and further studies are warranted to clarify this role.

In addition to TFRC, SHARPIN mRNA expression was also higher in various sarcoma cell lines, except VAESBJ, SW872 and HT1080 (Figure 3A). Furthermore, analysis of TCGA datasets indicated that high SHARPIN expression is significantly related to poor prognosis in all types of cancer, including sarcoma, suggesting the importance of SHARPIN in clinical practice. SHARPIN activates NF-κΒ, a well-known oncogenic factor. Although the role of SHARPIN in ferroptosis has not been elucidated fully, a previous study reported that it promotes the ubiquitination and degradation of p53, leading to upregulation of SLC7A11 and an increase in the resistance of cholangiocarcinoma cells to ferroptosis [57]. These findings contradict those of our current experiments, which suggest that SHARPIN reduces PGC1α and SLC7A11 expression and promotes the sensitivity of sarcoma cells to ferroptosis. We found that knockdown of SHARPIN reduced the expression levels of PRMT5 and NF-κΒ, and that inhibition of PRMT5 or NF-κΒ increased PGC1α and SLC7A11 expression in Yamato cells, suggesting that the effects of SHARPIN on PGC1α and SLC7A11 expression involve PRMT5 and NF-κΒ. In addition to PGC1α, NF-κΒ activation is necessary for RSL3-induced ferroptosis [25]. Another study reported that TNF-α and NF-κΒ are potentially involved in ferroptosis [58]. NF-κΒ can influence intercellular metabolism, including aerobic glycolysis and OXPHOS [59], and this regulation may be involved in SHARPIN-mediated regulation of ferroptosis sensitivity.

To our knowledge, no other reports have described the involvement of PRMT5 in ferroptosis. The function of PRMT5 in mitochondria has not been elucidated fully, although a previous study showed that inhibition of PRMT5 promotes mitochondrial function by increasing PPARα and PGC1α expression [60], which is consistent with our current findings. Moreover, there is functional crosstalk between PRMT5 and NF-κΒ via PRMT5-mediated methylation of arginine residues on NF-κΒ subunit p65 [61]. Furthermore, PRMT5 may epigenetically regulate gene expression through symmetric dimethylation of histones such as H3R8 and H4R3 [62]. Thus, PRMT5 may regulate ferroptosis sensitivity in multiple ways. Further research can reveal the precise mechanisms implicated in ferroptosis.

In terms of the PGC1α/NRF2/SLC7A11 axis, PGC1α can synergistically induce NRF2-mediated gene transcription through an increase in NRF2 activity [63]. In the present study, SHARPIN inhibition increased NRF2 protein. NRF2 protein expression is controlled via multiple regulatory mechanisms including transcriptional regulation, post-transcriptional regulation, and metabolic alteration etc. PGC1α can bind to the promoter of the NRF2 gene and transcriptionally induce NRF2 expression [64]. NRF2 is known as a critical antioxidant modulator and regulates SLC7A11 gene expression [65]. 

In addition to the PGC1α/NRF2/SLC7A11 axis, autophagy appears to be another mechanism underlying the ability of PGC1α to desensitize cells to ferroptosis downstream of SHARPIN. There was no correlation between the degree of autophagy and ferroptosis sensitivity among the cell lines used in this study (Figure 1C); however, autophagy impacts the degradation of ferritin, which increases ferroptosis due to an increase in the supply of ferrous iron [34]. Mitophagy, a mitochondria-specific autophagy, reduces the sensitivity of cells to ferroptosis [66]. A recent study found that BNIP3L/NIX-mediated mitophagy ameliorates oxidative stress [45]. Notably, this study demonstrated that inhibition of SHARPIN increases PGC1α-regulated BNIP3L/NIX-mediated mitophagy. 

BNIP3L/NIX is a mitochondrial outer membrane protein that belongs to the B-cell/CLL lymphoma (BCL) 2 family and was initially reported as a proapoptotic factor [67]. BNIP3L/NIX was also known to clear the mitochondria during the development of reticulocytes [68], that is, BNIP3L/NIX-mediated mitophagy. BNIP3L/NIX is transcriptionally induced by HIF1α, p53, and PGC1α [68]. In this study, BNIP3L/NIX-mediated mitophagy reduced ferroptosis sensitivity via induction of an antioxidant response, which is consistent with a previous report [45]. SHARPIN protects TNF-α-induced apoptosis via maintenance of mitochondrial function [69] whereas SHARPIN promotes RSL3-induced ferroptosis from our results. This paradoxical function of SHARPIN seems worth investigating in the future. 

Overall, our findings suggest that ferroptosis may be a promising therapeutic target in sarcoma, particularly in subpopulations with poor prognosis due to high TFRC and SHARPIN expression.

## 5. Conclusions

We found that TFRC and SHARPIN are related to poor overall survival according to the TCGA database. In addition, TFRC and SHARPIN were expressed at higher levels in sarcoma cell lines than in noncancer and carcinoma cell lines. SHARPIN promotes the sensitivity of sarcoma cells to ferroptosis. Further analyses revealed that the PGC1α/NRF2/SLC7A11 axis and BNIP3L/NIX-mediated mitophagy are regulated through NF-κΒ and PRMT5 downstream of SHARPIN. Our findings suggest that ferroptosis could have a therapeutic effect in sarcoma, particularly in subpopulations with high TFRC and SHARPIN expression.

## Figures and Tables

**Figure 1 cancers-15-03484-f001:**
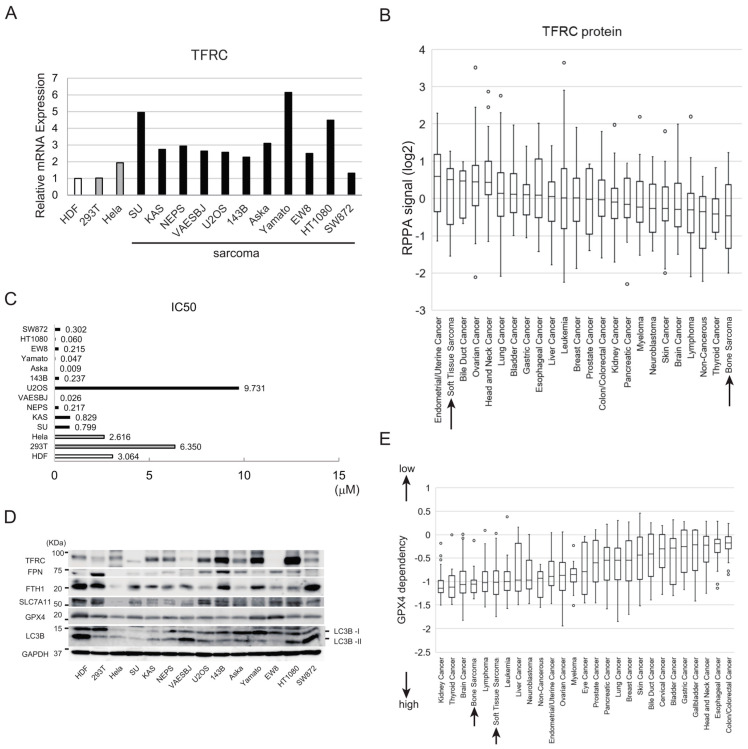
The sensitivities of noncancer, carcinoma, and sarcoma cell lines to ferroptosis. (**A**) Expression levels of the TFRC mRNA in 14 cell lines, including noncancer, carcinoma, and sarcoma cell lines, as determined by qPCR analysis. Expression levels of TFRC in each cell line are normalized to that in HDF. (**B**) Expression levels of the TFRC protein in diverse cell lines, using reverse-phase protein array deposited in the CCLE database. The y-axis is shown as a log2 scale. (**C**) IC50 values of RSL3 for noncancer, carcinoma, and sarcoma cell lines. (**D**) Immunoblot analyses of the expression levels of ferroptosis-related proteins (TFRC, FPN, FTH1, SLC7A11, GPX4, and LC3B) in the indicated cell lines. The uncropped blots are shown in Appendix A. (**E**) Analysis of the GPX4 dependency of cancer cells using Chronos, a dynamic model of CRISPR data (CCLE database).

**Figure 2 cancers-15-03484-f002:**
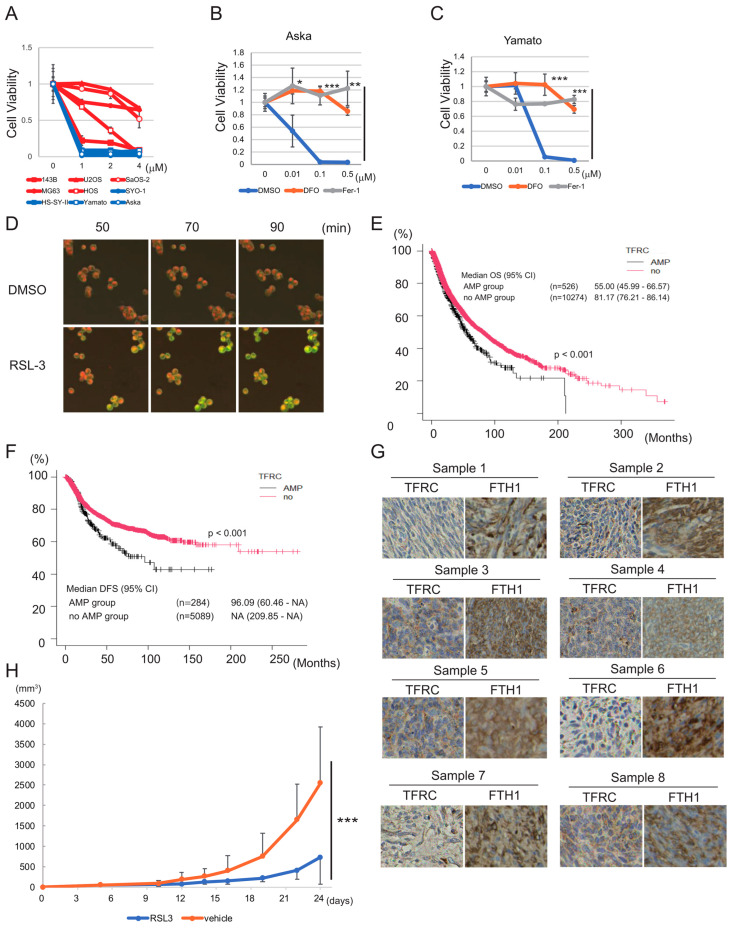
Synovial sarcoma cells are sensitive to ferroptosis inducers and express high levels of iron metabolism-related proteins. (**A**) Viability assays of five osteosarcoma cell lines (red lines) and four synovial sarcoma cell lines (blue lines) treated with RSL3. Cell viability is measured using the ratio of live cells in treated/control. (**B**,**C**) Viability assays of Aska (**B**) and Yamato (**C**) cells treated with RSL3 and with or without DFO or Fer-1. Cell viability is measured using the ratio of live cells in treated/control. (**D**) Lipid peroxidation induced by RSL3 in the Yamato cell line. Cells were pretreated with BODIPY 581/591 C11 and then treated with 1 μM RSL3 for the indicated time. (**E**,**F**) Kaplan–Meier curves showing the relationship between TFRC gene amplification (AMP) and OS (**E**) or DFS (**F**), based on a TCGA Pan-Cancer Atlas Studies dataset including all types of cancer ((**E**): n = 10,800; (**F**): n = 5373). Subjects were dichotomized into TFRC AMP and no AMP groups. (**G**) Immunostaining of synovial sarcoma clinical samples (n = 8) with anti-TFRC and anti-FTH1 antibodies. (**H**) In vivo experiment showing the effect of intratumoral injection of RSL3 on tumor volume in nude mice inoculated with 1.0 × 10^7^ Yamato cells. Inoculation was regarded as day 0. All quantitative data are presented as the mean ± SD. (**B**,**C**) Statistical significance was calculated via a one-way ANOVA. * *p* < 0.05; ** *p* < 0.005; *** *p* < 0.0005. (**E**,**F**) Statistical significance was calculated using a log-rank test. The *p*-value is indicated in each figure. (**H**) Statistical significance was calculated via a two-way ANOVA. *** *p* < 0.0005.

**Figure 3 cancers-15-03484-f003:**
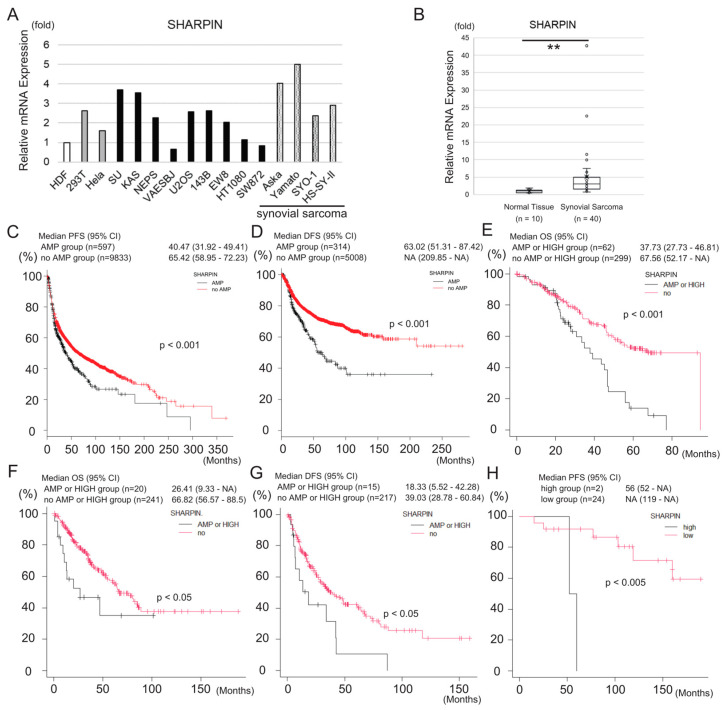
SHARPIN is expressed at a high level in synovial sarcoma cells and is associated with poor prognosis of the disease. (**A**) A qPCR analysis of SHARPIN mRNA expression in several sarcoma and non-sarcoma cell lines including four synovial sarcoma cell lines. (**B**) Box-and-whisker plot showing the results of a qPCR analysis of SHARPIN mRNA expression in clinical synovial sarcoma samples (n = 40) and normal tissues (n = 10). The expression levels of SHARPIN are normalized to those in normal tissues. Statistical significance was calculated using a Mann–Whitney U test. ** *p* < 0.005. (**C**,**D**) Kaplan–Meier curves showing the relationship between SHARPIN gene amplification (AMP) and PFS (**C**) or DFS (**D**), based on a TCGA Pan-Cancer Atlas Studies dataset including all types of cancer ((**C**): n = 10,430; (**D**): n = 5349). Subjects were dichotomized into SHARPIN AMP and no AMP groups. (**E**) Kaplan–Meier curve showing the relationship between OS and SHARPIN gene amplification (AMP) or mRNA expression, based on a ICGC/TCGA Pan-Cancer Analysis of Whole Genomes Consortium, 2020 (n = 362). Subjects were dichotomized into SHARPIN AMP or HIGH mRNA expression (z-score ≥ 2) and no AMP or LOW mRNA expression (z-score < 2) groups. (**F**,**G**) Kaplan–Meier curves showing the relationship between OS (**F**) or DFS (**G**) and SHARPIN gene amplification (AMP) or mRNA expression, based on a TCGA Firehose Legacy dataset of soft tissue sarcoma samples ((**F**): n = 261; (**G**): n = 232). Subjects were dichotomized into SHARPIN AMP or HIGH mRNA expression (z-score ≥ 2) and no AMP or LOW mRNA expression (z-score < 2) groups. (**H**) Kaplan–Meier curve showing the relationship between SHARPIN mRNA expression and PFS of patients with synovial sarcoma treated in our institute (n = 26). Subjects were dichotomized into SHARPIN high (z-score ≥ 2) and low (z-score < 2) groups. (**C**–**H**) Statistical significance was calculated using a log-rank test. The *p*-value is shown in each figure.

**Figure 4 cancers-15-03484-f004:**
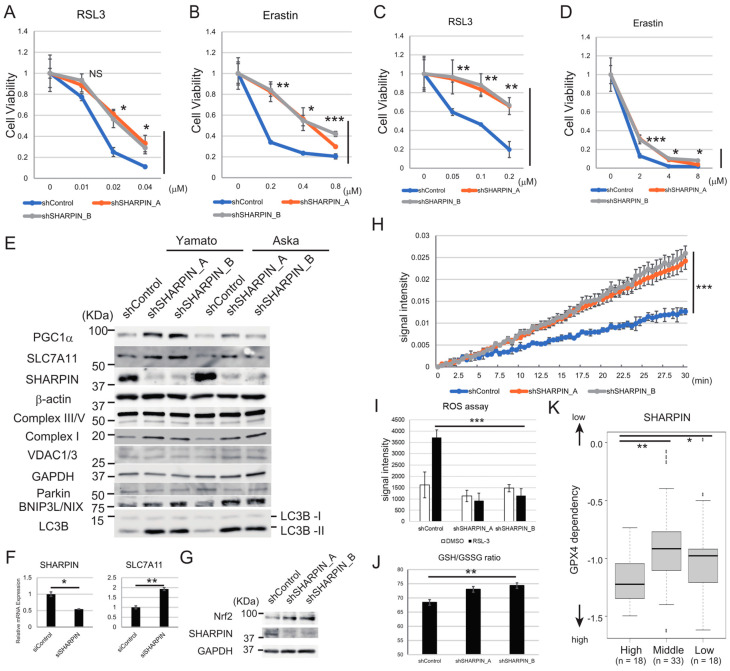
SHARPIN enhances the sensitivity of synovial sarcoma cell lines to ferroptosis via the PGC1α/SLC7A11 axis. (**A**–**D**) Viability assays of Aska (**A**,**B**) and Yamato (**C**,**D**) cells expressing scrambled or SHARPIN-specific shRNAs and treated with the indicated concentration of RSL3 (**A**,**C**) or erastin (**B**,**D**) for 24 h. (**E**) Immunoblot analyses of the effects of knockdown of SHARPIN on the expression levels of PGC1α, SLC7A11, SHARPIN, complex I, III, V, VDAC1/3, Parkin, BNIP3L/NIX, and LC3B in Yamato and Aska cells. (**F**) A qPCR analysis of the effect of transient SMARTpool siRNA-mediated knockdown of SHARPIN on SLC7A11 mRNA expression in Yamato cells. (**G**) Immunoblot analyses of the effects of knockdown of SHARPIN on the expression levels of NRF2 in Yamato cells. (**H**) Complex I activity in Yamato cells expressing scrambled or SHARPIN-specific shRNAs. Cells were seeded in identical numbers and incubated overnight. Signal intensity was then measured at the indicated time points. (**I**) ROS assay of Yamato cells expressing scrambled or SHARPIN-specific shRNAs. The cells were treated with or without 0.01 μM RSL3 for 24 h prior to the measurement of ROS activity. (**J**) GSH/GSSG ratio assay of Yamato cells expressing scrambled or SHARPIN-specific shRNAs. (**K**) Analysis of the relationship between SHARPIN mRNA expression levels and the GPX4 dependency of a bone and soft tissue sarcoma cohort using Chronos, a dynamic model of CRISPR data (CCLE database). The population below the first quantile (n = 18) was regarded as the low group, the population between the first and third quantile (n = 33) was regarded as the middle group, and the population above the third quantile (n = 18) was regarded as the high group. (**A**–**D**,**F**) Quantitative data are presented as the mean ± SD (n = 3). (**K**) A box-and-whisker plot is shown. (**A**–**D**,**I**,**J**) Statistical significance was calculated using one- or two-way ANOVA. * *p* < 0.05; ** *p* < 0.005; *** *p* < 0.0005; NS, not significant. (**F**) Statistical significance was calculated using a Student’s *t*-test. * *p* < 0.05; ** *p* < 0.005. (**K**) Statistical significance was calculated using a Kruskal–Wallis test and Bonferroni’s multiple comparison test. * *p* < 0.05; ** *p* < 0.005. The uncropped blots are shown in Appendix A.

**Figure 5 cancers-15-03484-f005:**
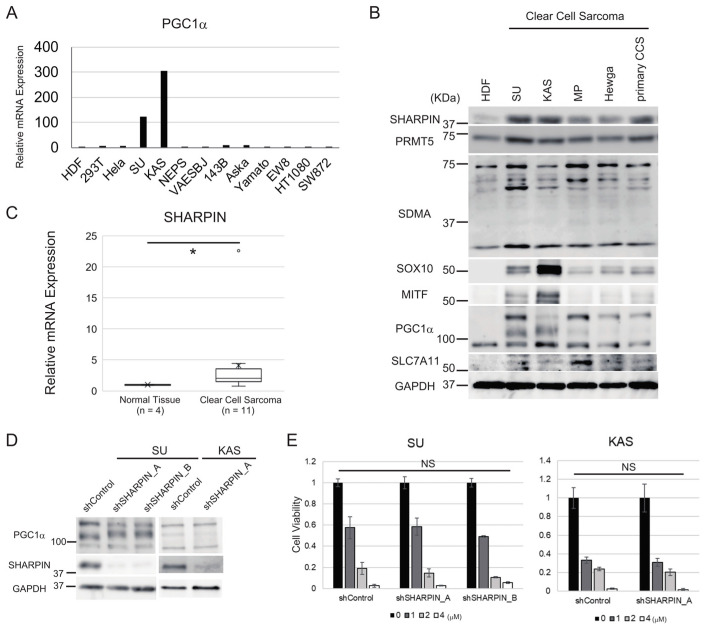
Aberrant PGC1α expression overwhelms the regulatory effect of SHARPIN inhibition on ferroptosis in CCS. (**A**) A qPCR analysis of PGC1α mRNA expression in several sarcoma or non-sarcoma cell lines including CCS cell lines (SU and KAS). (**B**) Immunoblot analyses of SHARPIN, PRMT5, SDMA, SOX10, MITF, PGC1α and SLC7A11 in four permanent CCS cell lines, one primary CCS cell line, and HDF. (**C**) A qPCR analysis of SHARPIN mRNA expression in CCS clinical samples (n = 11) and normal tissues (n = 4). (**D**) The effect of knockdown of SHARPIN on PGC1α protein expression in SU and KAS cell lines. (**E**) Viability assays of SU and KAS cells expressing scrambled or SHARPIN-specific shRNAs. The cells were treated with or without the indicated concentration of RSL3 for 24 h. Cell viability was measured using the ratio of live cells in treated/control. (**C**) A box-and-whisker plot is shown. Statistical significance was calculated using a Mann–Whitney U test. * *p* < 0.05. (**E**) Statistical significance was calculated via a one-way ANOVA or Student’s *t*-test. Quantitative data are presented as the mean ± SD (n = 3). NS, not significant. The uncropped blots are shown in Appendix A.

**Figure 6 cancers-15-03484-f006:**
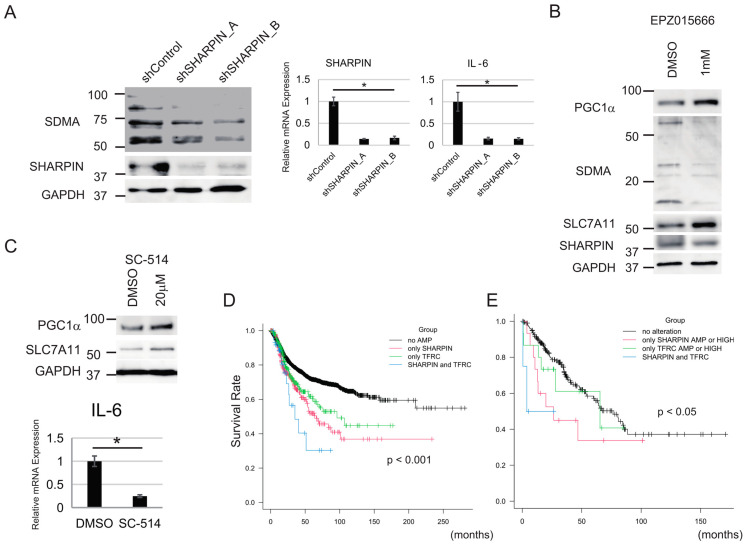
PRMT5 and NF-κΒ are essential regulators of ferroptosis downstream of SHARPIN. (**A**) Immunoblot and qPCR analyses of the effect of knockdown of SHARPIN on the expression levels of SDMA protein and IL-6 mRNA in Yamato cells. (**B**) The effect of a PRMT5 inhibitor, EPZ01566, on PGC1α, SLC7A11, and SDMA protein levels in Yamato cells. (**C**) The effect of a NF-κB inhibitor, SC-514, on PGC1α protein, SLC7A11 protein, and IL-6 mRNA levels in Yamato cells. (**D**) Kaplan–Meier curve showing the relationship between DFS and SHARPIN and/or TFRC gene amplification (AMP), based on a TCGA dataset of all types of cancer. Subjects were divided into SHARPIN only AMP, TFRC only AMP, SHARPIN and TFRC AMP, and no AMP groups. (**E**) Kaplan–Meier curve showing the relationship between OS and SHARPIN and/or TFRC gene amplification (AMP), based on a TCGA dataset of soft tissue sarcoma samples. Subjects were divided into SHARPIN only AMP or HIGH (z-score ≥ 2), TFRC only AMP or HIGH, SHARPIN and TFRC AMP or HIGH, and no AMP or HIGH groups. (**A**,**C**) Statistical significance was calculated via a one-way ANOVA or Student’s *t*-test. Quantitative data are presented as the mean ± SD (n = 3). * *p* < 0.05. (**D**,**E**) Statistical significance was calculated using a log-rank test. The *p*-value is shown in each figure. The uncropped blots are shown in Appendix A.

## Data Availability

The datasets generated and/or analyzed during this study are available from the corresponding author on request.

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
