# Peer review of "SHARPIN Enhances Ferroptosis in Synovial Sarcoma Cells via NF-κB- and PRMT5-Mediated PGC1α Reduction"

_cancers, 2023, doi:10.3390/cancers15133484_

Round 1

Reviewer 1 Report

In the manuscript entitled “SHARPIN enhances ferroptosis in synovial sarcoma cells via NF-KB- and PRMT5-mediated PGC1-É‘ reduction”, the authors support the role of ferroptosis as a possible therapeutic target in sarcoma, particularly in sub-populations with high TFRC and SHARPIN expression.

 The paper is well written and the search strategy and the study selection process are well designed. The manuscript has the potential to be accepted, but some major and minor points have to be clarified or fixed before.

-Major revisions:

1. Conclusion: this mandatory section is completely missed. The authors should include a better explanation of the importance of these new exciting results.

2. the Material and Methods section should include a short paragraph about the TGCA and CCLE in silica analysis.

3. Section results in Line 210: Although the authors argue the relationship between ferroptosis and autophagy in the discussion section, it should be helpful in the meaning of the showed results to introduce this relationship at this point, explaining the investigation of the LC3B marker.

4. In Fig 2 (A-B-C) what is the expression of cell viability? Abs, OD, Percentage... 

-Minor revisions:

In the materials and methods section:

1. line 75: the authors should explain the acronym of RSL3.

2. Line 84:define CCS.

3. Line 136: the number of cells should be corrected (5.0x103).

4. Lines 157 (Ros assay),162 (GSH/GSSG ration assay), and 166 (Statistics) should be separated as paragraphs (2.11, 2.12, 2.13)

5. The figure's legend should be separated from the main text.

6. In the results section, the titles of each paragraph should be highlighted (lines 227-228; 287-288;346-347;427-428; 474-475)

Author Response

Reviewer 1

In the manuscript entitled “SHARPIN enhances ferroptosis in synovial sarcoma cells via NF-KB- and PRMT5-mediated PGC1-É‘ reduction”, the authors support the role of ferroptosis as a possible therapeutic target in sarcoma, particularly in sub-populations with high TFRC and SHARPIN expression.

 The paper is well written and the search strategy and the study selection process are well designed. The manuscript has the potential to be accepted, but some major and minor points have to be clarified or fixed before.

  • Thank you for your review to improve the quality of our manuscript. We believe our manuscript is now hopefully acceptable for publication.

-Major revisions:

  1. Conclusion: this mandatory section is completely missed. The authors should include a better explanation of the importance of these new exciting results.

- We added conclusion as below. We are terribly sorry for forgetting conclusion. Line 504

We found that TFRC and SHARPIN are related to poor overall survival according to TCGA database. In addition, TFRC and SHARPIN were expressed at higher levels in sarcoma cell lines than in noncancer and carcinoma cell lines. SHARPIN promotes the sensitivity of sarcoma cells to ferroptosis. Further analyses revealed that the PGC1a/NRF2/SLC7A11 axis and BNIP3L/NIX-mediated mitophagy are regulated through NF-kB and PRMT5 downstream of SHARPIN. Our findings suggest that ferroptosis could have a therapeutic effect in sarcoma, particularly in subpopulations with high TFRC and SHARPIN expression.

  1. the Material and Methods section should include a short paragraph about the TGCA and CCLE in silica analysis.

-We added the following statement. Line 179

2.13. In silico analysis

All data from The Cancer Genome Atlas (TCGA) were obtained using cBioportal for

Cancer Genomics. TCGA database was analyzed according to a certain gene amplification and/or mRNA upregulation to clarify the association with survival. The Cancer Cell Line Encyclopedia (CCLE) was analyzed for RPPA signal and GPX4 dependency.

  1. Section results in Line 210: Although the authors argue the relationship between ferroptosis and autophagy in the discussion section, it should be helpful in the meaning of the showed results to introduce this relationship at this point, explaining the investigation of the LC3B marker.

-We added the following statement. Line 229

The LC3B, a marker of autophagic activity that is associated with ferroptosis sensitivity through ferritinophagy [34], varies depending on the cell lines; HDF, Aska, Yamato, HT1080 and SW872 cell lines highly expressed LC3B (Figure 1D).

  1. In Fig 2 (A-B-C) what is the expression of cell viability? Abs, OD, Percentage... 

-Cell viability is measured using the ratio of live cells in treated/control. We added this statement in overall figures of cell viability in figure legends.

-Minor revisions:

In the materials and methods section:

  1. line 75: the authors should explain the acronym of RSL3.

- RAS-selective lethal 3 (RSL3) was explained. Line 79

  1. Line 84:define CCS.

- CCS stands for clear cell sarcoma. Line 94

  1. Line 136: the number of cells should be corrected (5.0x103).

- 5.0 x 103 → 5.0 x 103  

We noticed the same mistakes in other sites. We corrected all the mistakes. Line 146

  1. Lines 157 (Ros assay),162 (GSH/GSSG ration assay), and 166 (Statistics) should be separated as paragraphs (2.11, 2.12, 2.13)

- We corrected the sentences as suggested. Line 168

  1. The figure's legend should be separated from the main text.

- We transferred figure legends next to each figure in the main text to the end of the manuscript after reference. Line 728

  1. In the results section, the titles of each paragraph should be highlighted (lines 227-228; 287-288;346-347;427-428; 474-475)

- We highlighted them using bold style. Line 238, 279, 313, 364, 396

Reviewer 2 Report

In the manuscript by Tamiya et al, the sensitivity of various sarcoma cell lines to ferroptosis was analysed, focusing on the role of SHARPIN, an activator of NF-kB.

A minor revision is needed here.

- L45-47 - So far, the role of ferroptosis in the development/treatment of various pathologies has been pointed out, so it is not clear why the authors mention only neurodegenerative diseases.

- What is the rationale for selecting the carcinoma cell lines used in the context of ferroptosis induction? Please add in the text.

- L81 - carcinoma and HDF cell line are missing

- Please add the proposed mechanisms of action of erastin as you did for RSL3.

- The "Conclusions" section is missing

- L232: the result for cell line 143B is missing

- Fig. 2D and G - the micrographs are too small

- please: check the use of abbreviations (PRMT5, CCS)

- please check the Greek letters in the text

- typing error: 107 instead of 107

Author Response

Reviewer 2

In the manuscript by Tamiya et al, the sensitivity of various sarcoma cell lines to ferroptosis was analysed, focusing on the role of SHARPIN, an activator of NF-kB.
-             Thank you for your review to improve the quality of our manuscript. We believe our manuscript is now hopefully acceptable for publication.

A minor revision is needed here.

- L45-47 - So far, the role of ferroptosis in the development/treatment of various pathologies has been pointed out, so it is not clear why the authors mention only neurodegenerative diseases.
- We changed the citation which includes cancer, cardiovascular diseases. Citation 10. Line 50

Han, C.; Liu, Y.; Dai, R.; Ismail, N.; Su, W.; Li, B. Ferroptosis and Its Potential Role in Human Diseases. Front Pharmacol 2020, 11, 1–19, doi:10.3389/fphar.2020.00239.

- What is the rationale for selecting the carcinoma cell lines used in the context of ferroptosis induction? Please add in the text.

-We added statement as below in material and methods section. Line 89
- Hela cell line is one of the most common carcinoma cell lines. HDF and HEK293T are also universal cell lines. Taking universality of these three cell lines into consideration, they can be regarded as representative non-sarcoma cell lines in the present study.   

- L81 - carcinoma and HDF cell line are missing
We added the following sentences. Line 89

HDF is an immortalized human dermal fibroblast cell line. HEK293T (293T) is Hu-man Embryonic Kidney cells. Hela is a cervical carcinoma cell line.

- Please add the proposed mechanisms of action of erastin as you did for RSL3.
Explanation was added as below. Line 213

Erastin, an inhibitor of the cystine-glutamate antiporter SLC7A11,

- The "Conclusions" section is missing

We added conclusion part. We are terribly sorry for missing an essential part. Line 504

We found that TFRC and SHARPIN are related to poor overall survival according to TCGA database. In addition, TFRC and SHARPIN were expressed at higher levels in sarcoma cell lines than in noncancer and carcinoma cell lines. SHARPIN promotes the sensitivity of sarcoma cells to ferroptosis. Further analyses revealed that the PGC1a/NRF2/SLC7A11 axis and BNIP3L/NIX-mediated mitophagy are regulated through NF-kB and PRMT5 downstream of SHARPIN. Our findings suggest that ferroptosis could have a therapeutic effect in sarcoma, particularly in subpopulations with high TFRC and SHARPIN expression.

- L232: the result for cell line 143B is missing
-143B 0.237 mM was added. Line 244

- Fig. 2D and G - the micrographs are too small
Figure 2 was changed to enlarge the images of 2D and 2G. Line 276

- please: check the use of abbreviations (PRMT5, CCS)
-Original name was added in Simple Summary. Line 17

nuclear factor-kappa B (NF-kB) and protein arginine methyltransferase 5 (PRMT5)

CCS was also explained in cell culture in material and methods section. Line 94

SU-CCS-1 (SU), KAS, MP-CCS-SY (MP), and Hewga-CCS (Hewga) are clear cell sar-coma (CCS) cell lines.

- please check the Greek letters in the text
We checked symbols and corrected them to right style.

- typing error: 107 instead of 107

We noticed the same mistakes in other sites. We corrected all the mistakes.

Reviewer 3 Report

This study by Hironari et al., investigated the oncogenic factors SHARPIN enhances ferroptosis in sarcoma. The study is well-designed and has merits, however, the authors should consider below mentioned critique that could make the study more appreciable. 

1)      Animal studies (line number 73) section 2.1 should be provide graphical chat, readers can easily understand

2)      In figure 1 D SLC7A11 expression in 293T cells not clearly, if possible provide clear image.

3)      There is a missing conclusion (line number 579) that needs to be written

Moderate editing of English language required

Author Response

Reviewer 3

This study by Hironari et al., investigated the oncogenic factors SHARPIN enhances ferroptosis in sarcoma. The study is well-designed and has merits, however, the authors should consider below mentioned critique that could make the study more appreciable. 

 -Thank you for your review to improve the quality of our manuscript. We believe our manuscript is now hopefully acceptable for publication.

  • Animal studies (line number 73) section 2.1 should be provide graphical chat, readers can easily understand

Experimental scheme was added as below. Line 85

  • In figure 1 D SLC7A11 expression in 293T cells not clearly, if possible provide clear image.

Contrast was adjusted. SLC7A11 is now clearly recognized. Line 235

3)      There is a missing conclusion (line number 579) that needs to be written

We added conclusion part. We are terribly sorry for missing an essential part. Line 504

We found that TFRC and SHARPIN are related to poor overall survival according to TCGA database. In addition, TFRC and SHARPIN were expressed at higher levels in sarcoma cell lines than in noncancer and carcinoma cell lines. SHARPIN promotes the sensitivity of sarcoma cells to ferroptosis. Further analyses revealed that the PGC1a/NRF2/SLC7A11 axis and BNIP3L/NIX-mediated mitophagy are regulated through NF-kB and PRMT5 downstream of SHARPIN. Our findings suggest that ferroptosis could have a therapeutic effect in sarcoma, particularly in subpopulations with high TFRC and SHARPIN expression.

Reviewer 4 Report

Dear Editor,

 I have read the manuscript titled "SHARPIN enhances ferroptosis in synovial sarcoma cells via 2 NF-ĸB- and PRMT5-mediated PGC1α reduction". The aim of the study, as stated by the authors, was to investigate the impact of ferroptosis on sarcoma. The findings suggest that SHARPIN expression is strongly linked to reduced survival rates in cancer patients, including those with sarcoma. Additionally, SHARPIN enhances the sensitivity of sarcoma cells to ferroptosis. The authors conclude that ferroptosis could potentially serve as a therapeutic intervention for sarcoma, especially in subgroups with high TFRC and SHARPIN expression levels.

The paper's results are commendable, and the employed procedures are well-executed. The manuscript is well-organized, and the results substantiate the discussion. Regrettably, a conclusion section is missing from the manuscript.

To move forward with publication, the author needs to address the following issues:

1. The conclusion section needs to be included.

2. Given the number of experiments and results, the discussion section must be expanded. Consider developing the discussion section further or combining it with the results section.

3. The size of the figures is excessive and requires better balance with the essential information. For instance, Figure 6F should be excluded as it serves more as a graphical abstract or summary. The authors should strive to minimize the amount and variety of information in each figure and keep additional details in the supplementary information.

4. The section on results may take some time to follow due to the abundance of details. I think you should make the section easy to read.

The manuscript is well-written in good-quality English language. However, the section on results may take some time to follow due to the abundance of details. I suggest making the section easy to read.

Author Response

Reviewer 4

 I have read the manuscript titled "SHARPIN enhances ferroptosis in synovial sarcoma cells via 2 NF-ĸB- and PRMT5-mediated PGC1α reduction". The aim of the study, as stated by the authors, was to investigate the impact of ferroptosis on sarcoma. The findings suggest that SHARPIN expression is strongly linked to reduced survival rates in cancer patients, including those with sarcoma. Additionally, SHARPIN enhances the sensitivity of sarcoma cells to ferroptosis. The authors conclude that ferroptosis could potentially serve as a therapeutic intervention for sarcoma, especially in subgroups with high TFRC and SHARPIN expression levels.

The paper's results are commendable, and the employed procedures are well-executed. The manuscript is well-organized, and the results substantiate the discussion. Regrettably, a conclusion section is missing from the manuscript.

 -Thank you for your review to improve the quality of our manuscript. We believe our manuscript is now hopefully acceptable for publication.

To move forward with publication, the author needs to address the following issues:

  1. The conclusion section needs to be included.

We added conclusion part. We are terribly sorry for missing an essential part. Line 504

We found that TFRC and SHARPIN are related to poor overall survival according to TCGA database. In addition, TFRC and SHARPIN were expressed at higher levels in sarcoma cell lines than in noncancer and carcinoma cell lines. SHARPIN promotes the sensitivity of sarcoma cells to ferroptosis. Further analyses revealed that the PGC1a/NRF2/SLC7A11 axis and BNIP3L/NIX-mediated mitophagy are regulated through NF-kB and PRMT5 downstream of SHARPIN. Our findings suggest that ferroptosis could have a therapeutic effect in sarcoma, particularly in subpopulations with high TFRC and SHARPIN expression.

  1. Given the number of experiments and results, the discussion section must be expanded. Consider developing the discussion section further or combining it with the results section.

-Exactly. We discussed more as below. Line 465

To our knowledge, no other reports have described the involvement of PRMT5 in ferroptosis. The function of PRMT5 in mitochondria has been not elucidated fully, although a previous study showed that inhibition of PRMT5 promotes mitochondrial function by increasing PPARa and PGC1a expression [60], which is consistent with our current findings. Moreover, there is functional crosstalk between PRMT5 and NF-kB via PRMT5-mediated methylation of arginine residue on NF-kB subunit p65 [61]. Furthermore, PRMT5 may epigenetically regulate gene expression through sym-metric dimethylation of histones such as H3R8 and H4R3 [62]. Thus, PRMT5 may reg-ulate ferroptosis sensitivity in multiple ways. Further research can reveal the precise mechanisms implicated in ferroptosis.

In terms of PGC1a/NRF2/SLC7A11 axis, PGC1a can synergistically induce NRF2-mediated gene transcription through increase in NRF2 activity [63]. In the pre-sent study, SHARPIN inhibition increased NRF2 protein. NRF2 protein expression is controlled via multiple regulatory mechanisms including transcriptional regulation, post-transcriptional regulation, and metabolic alteration etc. PGC1a can bind to pro-moter of NRF2 gene and transcriptionally induce NRF2 expression [64]. NRF2 is known as a critical antioxidant modulator and regulates SLC7A11 gene expression [65].   

In addition to the PGC1a/NRF2/SLC7A11 axis, autophagy appears to be another mechanism underlying the ability of PGC1a to desensitize cells to ferroptosis down-stream of SHARPIN. There was no correlation between the degree of autophagy and ferroptosis sensitivity among the cell lines used in this study (Figure 1C); however, autophagy impacts the degradation of ferritin, which increases ferroptosis due to an in-crease in the supply of ferrous iron [34]. Mitophagy, a mitochondria-specific autopha-gy, reduces the sensitivity of cells to ferroptosis [66]. A recent study found that BNIP3L/NIX-mediated mitophagy ameliorates oxidative stress [67]. Notably, this study demonstrated that inhibition of SHARPIN increases PGC1a-regulated BNIP3L/NIX-mediated mitophagy.

BNIP3L/NIX is a mitochondrial outer membrane protein that belongs to B-cell/CLL lymphoma (BCL) 2 family and was initially reported as a proapoptotic fac-tor [68]. BNIP3L/NIX was also known to clear the mitochondria during the develop-ment of reticulocytes [69], that is, BNIP3L/NIX-mediated mitophagy. BNIP3L/NIX is transcriptionally induced by HIF1a, p53, and PGC1a [69]. In this study, BNIP3L/NIX-mediated mitophagy reduced ferroptosis sensitivity via induction of an-tioxidant response, which is consistent with a previous report [67]. SHARPIN protects TNF-a-induced apoptosis via maintenance of mitochondrial function [70] whereas SHARPIN promotes RSL3-induced ferroptosis from our results. This paradoxical func-tion of SHARPIN seems worth investigating in the future.  

  1. The size of the figures is excessive and requires better balance with the essential information. For instance, Figure 6F should be excluded as it serves more as a graphical abstract or summary. The authors should strive to minimize the amount and variety of information in each figure and keep additional details in the supplementary information.

-As reviewer 4 and other reviewers also pointed out, we changed Fig1, Fig2 as well. Fig6F was excluded from Fig6. Graphical abstract was added as below

  1. The section on results may take some time to follow due to the abundance of details. I think you should make the section easy to read.

- We transferred figure legends to the end of the manuscript after reference. Moreover, the titles of each paragraph should be highlighted using bold style. And as the reviewer kindly indicated, some Figures were arranged so that readers can more easily understand the manuscript.

The manuscript is well-written in good-quality English language. However, the section on results may take some time to follow due to the abundance of details. I suggest making the section easy to read.

Round 2

Reviewer 1 Report

The authors improved the manuscript according to the suggestions and now it can be accepted in this form. 

Reviewer 4 Report

Dear Editor,

 I have read the improved manuscript titled "SHARPIN enhances ferroptosis in synovial sarcoma cells via 2 NF-ĸB- and PRMT5-mediated PGC1α reduction". The authors have addressed many of the comments I made. I consider that the paper can be accepted for publication according to its scientific meaning.

Regarding the form, the only aspect that the authors needed to address fully is the number of images included in each figure. Much of that information can be provided as supplementary information. The authors must leave just the essentials. For example, in Figures 2D and 2G is tough to appreciate the staining color o cell shape due to the size of the images. I don't consider relocating the footnote of each figure to the end of the text as an optimal solution.

Minor editing of English language required